# Interactive Task Planning with Language Models

**Boyi Li\***        *boyili@berkeley.edu*
*UC Berkeley*

**Philipp Wu\***        *philippwu@berkeley.edu*
*UC Berkeley*

**Pieter Abbeel**        *pabbeel@berkeley.edu*
*UC Berkeley*

**Jitendra Malik**        *malik@eecs.berkeley.edu*
*UC Berkeley*

**Reviewed on OpenReview:** *https://openreview.net/forum?id=GAhGMttRIo*

## Abstract

An interactive robot framework accomplishes long-horizon task planning and can easily generalize to new goals and distinct tasks, even during execution. However, most traditional methods require predefined module design, making it hard to generalize to different goals. Recent large language model based approaches can allow for more open-ended planning but often require heavy prompt engineering or domain specific pretrained models. To tackle this, we propose a simple framework that achieves interactive task planning with language models by incorporating both high-level planning and low-level skill execution through function calling, leveraging pretrained vision models to ground the scene in language. We verify the robustness of our system on the real world task of making milk tea drinks. Our system is able to generate novel high-level instructions for unseen objectives and successfully accomplishes user tasks. Furthermore, when the user sends a new request, our system is able to replan accordingly with precision based on the new request, task guidelines and previously executed steps. Our approach is easy to adapt to different tasks by simply substituting the task guidelines, without the need for additional complex prompt engineering. Please check more details on our **Project Page** and **Demo Video**.

## 1 INTRODUCTION

The rise of Large Language Models (LLMs) and proliferation of chatbots highlight the importance of human interaction in AI systems. Beyond merely executing user commands, an autonomous agent should fluidly receive and incorporate feedback at any step during the execution process. Consider the seemingly straightforward human task of preparing a flavorful milk tea drink, which we study in this work. Such a task, while simple to humans, requires a robot agent to decompose it into numerous intermediate steps. Not only does the robot need to generate and execute the steps precisely, but the robot should also remain receptive to real-time modifications or feedback to the initial request. For example, the user might request some boba to be added to their drink. A robot should be able to seamlessly incorporate such interaction during operation.

In light of these challenges, we propose a simple framework for **I**nteractive **T**ask **P**lanning with language models, denoted as ITP. Our framework leverages LLMs to plan, execute, and adapt to user inputs throughout the task lifecycle. Figure 1 illustrates an exemplary interaction with our system. Our primary objective is to offer a blueprint for deploying real-world robotic systems that harness pretrain language models to coordinate the execution of lower-level skills of a robot in a simple manner.

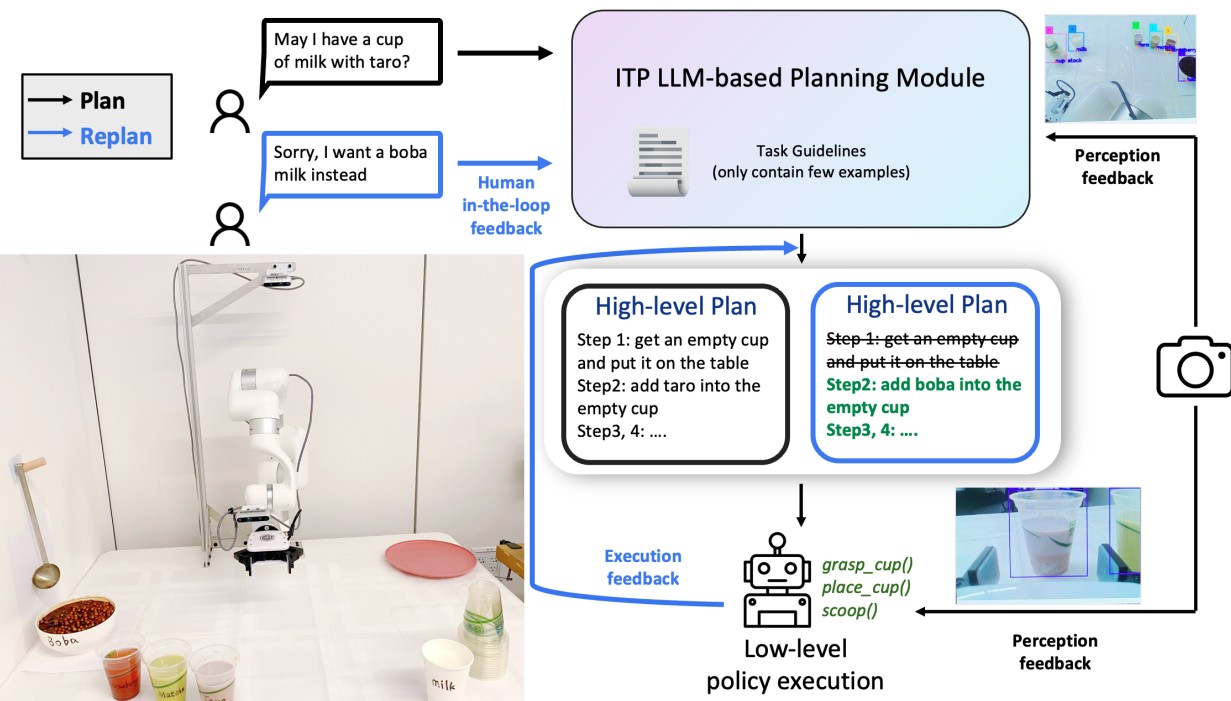

Figure 1: An overview of ITP. ITP generates high-level plans and executes the low-level robot skills through LLMs. Our system generates a high-level plan based on user requests and task guidelines. When the system is interrupted with a new request, the system will replan, taking into consideration prior completed steps and task guidelines. Each step of the plan is executed by leveraging an LLM executor, equipped with additional visual grounded outputs, to call lower-level skills. In the example shown, the user first requests '*May I have a cup of milk with taro?*', a request for which the high-level plan is not provided in the task guidelines. After the robot has finished the first step, the user wants to revise the order to a boba milk. Our system is able to replan and make a new set of high-level steps based on the new request, a history of completed steps, and task guidelines, which can then be completed by the lower level execution module.

In this work, we utilize GPT-4 (OpenAI, 2023) as the language model backbone. ITP consists of two primary modules. First, a high level planner, which takes as input a prompt and a user request to specify the task and outputs a step by step plan. Second, a low level executor, which tries to achieve a given step by converting robots skills into a functional API, which enables GPT-4's function calling capabilities to directly interact with the robot, abstracting code level details from the system. ITP does not require the training of additional value functions such as SayCan (Zeng et al., 2022; Huang et al., 2023), and does not require code level prompts such as Code as Policies(Singh et al., 2023) or ProgPrompt (Liang et al., 2022). Furthermore, ITP dynamically generates novel plans and re-adjusts its plan based on user input. We hope our framework will be useful for accomplishing a wide range of interactive robot tasks and will release our codebase to foster advancements in this field. We outline the key features of ITP below:

1. ITP is a training-free robotic system for interactive task planning with language models with a focus on simplicity. We showcase ITP in the context of a real-world boba drink-making robot that integrates planning, vision and skill execution.

2. ITP leverages a simple prompt format, which we show is effective across simulated and real settings. Additionally, our ITP system converts the lower-level skills into a functional language-based API that can be leveraged by any function calling LLM (ie. GPT-4). This enables a user to prompt the system through natural language rather than code, removing the need for code-level prompt engineering.

3. Our system exhibits robustness in adapting to user requests during execution, allowing it to consider the updated goals, previously completed steps, and task guidelines in order to replan new steps.

# 2 RELATED WORK

## 2.1 Task planning

Task planning, the problem of developing a plan to achieve a desired goal, is an integral component of our work. Traditionally, task planning in the robotics commonly leverages symbolic planners which reduces the planning problem into a search problem (Ghallab et al., 2004; Bonet & Geffner, 2001). Practitioners often define the problem in a declarative language (Jiang et al., 2019; Ghallab et al., 1998; Lifschitz, 2008; Fikes & Nilsson, 1971), which can be restrictive as it requires meticulous definitions of the problem parameters, such as actions, their preconditions and their effects. Task and motion planning (TAMP), takes task planning a step further and also jointly considers the lower level execution during higher level planning (Garrett et al., 2020a; Mansouri et al., 2021). TAMP methods also consider symbolic representations and leverage search algorithms to extract the final sequence of lower-level primitives and has seen success in robotic manipulation (Siméon et al., 2004; Garrett et al., 2017; 2020b). As the search space can often be prohibitively large, some methods leverage hierarchy and/or sampling (Bacchus & Yang, 1991; Plaku & Hager, 2010; Kaelbling & Lozano-Pérez, 2011; Kaelbling & Lozano-Pérez, 2013). Our approach replaces traditional planning pipelines with LLMs, offering common-sense reasoning, enhanced interaction capabilities, and the ability to define the problem's scope using natural language.

## 2.2 Language Models as Planners in Robotics

Due to the popularity of LLMs, there has been a rising interest in leveraging LLMs as a policy in robot systems. One work in this direction leverages LLMs as zero-shot planners in simulated embodied settings (Huang et al., 2022) by converting the scene and task definitions into language, then letting the LLM directly predict actions. Works such as Socratic models, SayCan, and Grounded decoding (Zeng et al., 2022; Ahn et al., 2022; Huang et al., 2023) follow in this line of work, coordinating many large pre-trained models with a robot to solve various tasks. In contrast to approaches like SayCan (Zeng et al., 2022), which necessitate a pretrained value function to ground actions, we rely on prompting the language model with task guidelines and robot skills. This implicitly encodes preconditions and effects, reminiscent of traditional declarative task planning approaches but can be done so with natural language, which is more expressive and easier for the average user to tune. G-PlanET (Lin et al., 2023) explores using language models for robot task planning by generating high-level subgoals in simulated environments. Tidy bot (Wu et al., 2023) shows that LLMs can help a robot follow a user's preferences based on a few examples. We also prompt the model with a small set of examples but explore generalization to new goals. Reflect (Liu et al., 2023b) uses large models to make an agent recount their experiences and correct failures. LLMs have also been used to allow robots to seek help when uncertain (Ren et al., 2023).

A related approach, used in Code as Policies (Liang et al., 2022) and ProgPrompt (Singh et al., 2023), leverages the code writing capabilities of LLMs to generate code that a robot agent can execute directly. This often requires heavy prompt engineering of example code to show the model how to properly use the provided functions to accomplish a directive. Language-guided Robot Skill Learning (Ha et al., 2023), like us, takes a hierarchical approach to LLM planning, but assumes access to the simulator which provides ground truth state information. Voyager (Wang et al., 2023) uses LLMs to build a lifelong learning agent for Minecraft by having the agent explore and solve new tasks through writing code that interacts with the API.

Our work falls into this general category of leveraging LLMs to plan, and then execute actions in the environment. In contrast to prior work, we allow the LLM to generate a high-level plan based on contextual information. These low-level plans are then executed directly by an LLM with access to the functional API of the robot using a pre-trained VLM to ground the visual scene into primitives. Our work focuses on how to instantiate such a system in the real world.

```
Options:
Pure milk, Strawberry milk, Boba milk
Instructions:
Pure milk
Material:  milk
Steps:
0) get an empty cup and bring it to the working area
1) pour the milk into the working cup
2) put the working cup in the finished location
Strawberry milk
Material:  strawberry jam, milk
Steps:
0) get an empty cup and bring it to the working area
1) add strawberry jam to the working cup
2) pour the milk into the working cup
3) put the working cup in the finished location
Boba milk
Material:  boba, milk
Steps:
0) get an empty cup and bring it to the working area
1) add boba to the working cup
2) pour the milk into the working cup
3) put the working cup in the finished location
Available material we have now:
boba, strawberry jam, mango jam, matcha powder, taro, milk,blueberry
```

Task Guidelines 1: The task guidelines we use for our drink making experiments. Task guidelines only need contain simple; human interpretable few shot examples and a description of relevant assets in the scene.

## 3 METHOD

ITP offers a blend of high-level planning and low-level execution, powered by LLMs. In contrast to prior work (Liang et al., 2022; Singh et al., 2023), our approach enables the LLM to create a high-level plan informed by contextual information in the form of a list of steps. Each step of this plan is subsequently realized by another LLM with access to the functional API of the robot. A pre-trained VLM grounds the visual scene into language. Our work focuses on how to instantiate such a system in the real world. Our framework, shown in Figure 1, with a more detailed breakdown in Figure 3, consists of a hierarchy of two levels, the high level and low level.

### 3.1 High-level Planning

**LLMs for Planning.** We utilize GPT-4 (OpenAI, 2023) as our language model, one of the most capable LLMs available at the time of this writing. The high-level planner takes as input a given prompt, task guidelines, and a user request, and outputs a step-by-step plan to execute the request. It also retains past user interactions for any necessary replanning.

**Simple Prompting Strategy.** Task guidelines, described using natural language, outline the scope of the robot's tasks and are provided to the high-level planner. The prompt contains the user's request and task guidelines which contain few-shot prompt examples of plans in the given domain of interest, and a description of the available materials to the robot. In our milk tea system, the task guidelines consist of a select set of menu items, their corresponding preparation steps, and a list of relevant ingredients. This includes the procedures for a few drinks like 'pure milk' and 'boba milk'. See Task Guidelines 1 for the exact task guidelines we used in our experiments. Our system utilizes these guidelines to determine the feasibility of making a new drink based on available materials. Leveraging LLMs' few-shot learning capabilities (Brown et al., 2020), ITP can generalize from the baseline guidelines to make detailed steps for other drinks such as 'boba strawberry milk' or 'taro milk'.

### 3.2 Low-level Execution

The low-level executor takes each generated step and does its best to complete it successfully, conditioned on additional information about the scene and available robot skills. We use pretrained vision modules to convert the scene into a language compatible format. Additionally, we translate robot skills into a functional

API, automatically translating the python docstring of the robot skills into callable functions by the language model.

**Visual Scene Grounding.** The role of the vision module in our system is to process the camera inputs into concise language descriptions of the scene, which can further be processed for planning and task execution downstream. In our drink-making system, the visual grounding system accepts a list of menu items and generates corresponding bounding boxes. Using a simple projective mapping, we then approximate the $x$ and $y$ locations of each item in the robot frame. We employ the pretrained VLM: Grounded-DINO (Liu et al., 2023a), a variant of the original DINO model (Caron et al., 2021) fine-tuned for extracting 2D bounding boxes given language descriptions. The final text description is represented as a dictionary of object description to $(x, y)$ location. The vision system gives a holistic 'understanding' of the scene, despite the location assignments being imprecise.

**Robot Skill Grounding.** The language model interfaces with a predefined skill set in Python that controls the robot. These skills are translated into a functional API by parsing of function definitions and related doc strings. This can be directly used with GPT's function-calling layer (OpenAI, 2023). In contrast to methods like ProgPrompt or Code as Policies, our system does not require examples or function internals when prompting the LLM. Instead, more detailed prompting of the language model can be specified via natural language in the documentation of the functions.

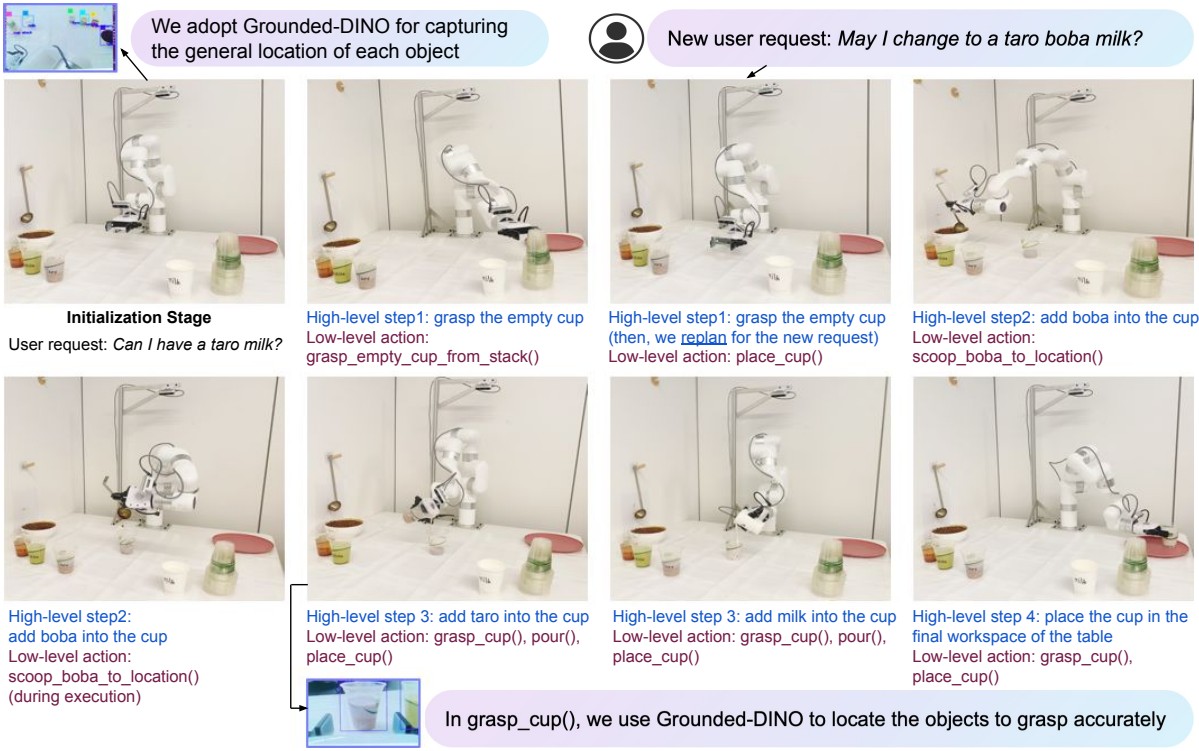

Figure 2: An example of ITP to make a cup of taro milk with boba. Our system first makes a high-level plan based on the users request using GPT-4: step 1) grasp the empty cup, step 2) add taro into the cup, step 3) add milk into the cup, step 4) place the cup in the final workspace. For each step in the high-level plan, we feed step into another instance of GPT-4 and obtain the corresponding low-level actions which is directly executed on the robot. As for the perception component, ITP uses Grounded-DINO to capture the general location of each object and locate the object accurately when taking the actions. However, after grasping the empty cup, the user sends a new request '*May I change to a taro boba milk?*'. Considering the history of completed steps, the system replans and generates the following high-level steps and low-level executions. The following plan has been changed to: step 2) add boba into the cup, step 3) add taro into the cup, step 4) add milk into the cup, step 5) place the cup in the final workspace.

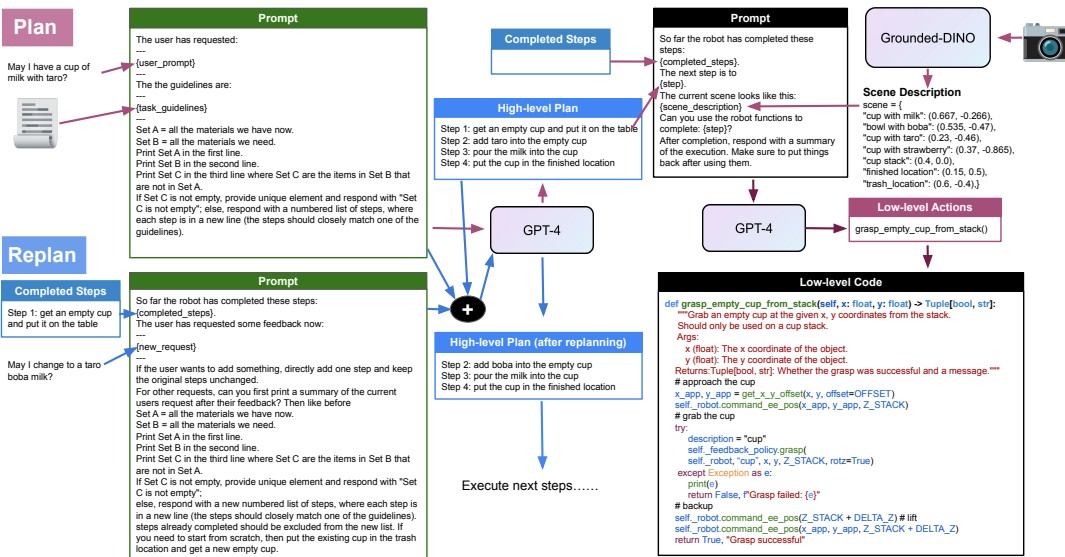

Figure 3: Detailed diagram of ITP with specific examples of user requests and task guidelines and replanning. During "*Plan*": we feed user requests and task guidelines to complete the prompt and input it into GPT-4 to obtain a high-level plan. We input the history of completed steps and next step to prompt into the lower level executor to call the corresponding low-level actions. Once the lower level executor completes a step, we will maintain the history by storing it into *Completed Steps*. GPT-4 directly makes function calls to a predefined robot skill library (which could be learned or handcrafted). During "*Replan*": we feed the completed steps and new request to create a new prompt, we append this new prompt to the previous conversation context and input the whole message into GPT-4 to obtain a new high-level plan. We refer this procedure as replanning, which previous language-based task planning methods have not considered. The low level executor then completes the next steps based on the new high-level plan.

### 3.3 Replanning

Beyond the aforementioned components, ITP considers new requests from the user as *human-in-the-loop feedback*. We allow a human to interpret the robot execution at any stage with a new prompt. This then triggers our replanning pipeline. The system will consider completed steps, task guidelines, the new request, and the chat history to generate a new plan. The details of replanning are see in Figure 3. We also showcase ITP's adeptness in planning and adaptive replanning of the same example in Figure 2.

## 4 EXPERIMENTS

### 4.1 Robot Experiments

In our experiments, we focus on a drink-making system. Within the given scene, the robot is supplied with a set of ingredients that it must combine to produce a specific drink. Our setup also has includes an overhead camera that captures image of the scene. We leverage Grounded-DINO to process the overhead camera images for open vocabulary object detection and scene understanding.

For the robot, we provide a predefined set of skills, which include actions like "grasp_cup", "pour", and "scoop_boba_to_location". The "grasp_cup" skill is implemented with a feedback policy that centers the gripper on the cup, given the approximate location from the scene description, enabling the robot to grasp it reliably. The "pour" skill is designed to accept a location and a descriptive cue of the ingredient being poured. This level of specification enables milk to be poured more than specific flavors. For example, when making a matcha latte, the *pour* function is provided "*matcha*" or "*milk*" as input. When the input is "*matcha*", the controllable tilt angle will be small, while when the input is "*milk*", the controllable tilt angle will be much larger. This ensures that the robot can pour more milk and a bit of matcha liquid.

| User Request | Difficulty Level | Code as Policies | | ITP | |
|---|---|---|---|---|---|
| | | High-level Planning | Success | High-level Planning | Success |
| *I would like to order a cup of milk.* | Existed | 3/3 | ✓ | 3/3 | ✓ |
| *I want to order a boba milk.* | Existed | 2/4 | ✗ | 4/4 | ✓ |
| *Can I have a cup of strawberry milk?* | Existed | 4/4 | ✓ | 4/4 | ✓ |
| *I want a matcha latte.* | Zero-shot easy | 4/4 | ✓ | 4/4 | ✓ |
| *May I have a cup of milk with taro?* | Zero-shot easy | 3/3 | ✓ | 3/3 | ✓ |
| *I want taro milk with boba.* | Zero-shot moderate | 3/5 | ✗ | 5/5 | ✓ |
| *Can I get a strawberry boba milk?* | Zero-shot moderate | 3/5 | ✗ | 5/5 | ✓ |
| *I want to order a strawberry matcha milk.* | Zero-shot moderate | 5/5 | ✓ | 5/5 | ✓ |
| *I'd order a strawberry matcha milk with boba.* | Zero-shot hard | 3/6 | ✗ | 6/6 | ✓ |
| *I would like a cup of passion fruit milk.* | Unavailable material | - | ✗ | - | ✓ |
| Total | - | 80% | 5/10 | 100% | 10/10 |

Table 1: Quantitative results with real robots for high-level planning rate and success rate with various user requests. For high-level planning, we extract planning accuracy by dividing the number of successful steps by the total number of steps, shown as '*Successful Steps / Total Steps*'. We determine success by whether the robot successfully accomplishes the task. To calculate the overall high-level planning score, we average the performance across all user requests.

## 4.2 Comparison on Task Planning

We consider Code as Policies as a baseline. Code as Policies provides a formulation for language model-generated programs executed on real systems by prompting a text completion model with code examples. For a fair comparison, not only do we provide Code as Policies with the same information as given in ITP in the form of comments, but we also provide an additional 40 lines of code prompts providing example usage, as is done in Code as Policies. For both ITP and Code as Policies, we provide user requests and task guidelines as inputs. The task guidelines include 3 instances, along with their associated high-level planning steps, current available material, and other task-specific conditions. Our task guidelines are shown in Task Guidelines 1.

We evaluate the methods on two criteria: the number of high-level steps correctly generated and whether the real robot successfully finished the task. We send user requests of varying complexity levels, including 'existed', 'zero-shot easy,' 'zero-shot moderate', 'zero-shot hard' and 'unavailable material'. 'Zero-shot' means the instruction for making the corresponding drink is not provided in the task guidelines. 'Unavailable' indicates that we do not have the material for the requested beverage. We show the results in Table 1. We could notice that ITP is robust in high-level plan generation and can easily be generalized to novel instructions of unseen drinks or unavailable drinks. For example, the user sends the request '*I would like a cup of passion fruit milk.*' However, passion fruit jam is not available, so the system will provide the response '*Passion fruit jam is not available*' and stop the program. In comparison, Code as Policies failed to achieve this objective. To understand the failure case of Code as Policies, we provide some observations: 1) when making a cup of milk with boba, the system attempted to scoop boba from the working cup, improperly adhering to the correct usage of the lower-level skill. 2) When the prompt is more complex (9th row), the system adds milk first and then adds the boba, resulting in an incorrect execution order. 3) When the material is not available, it cannot justify that passion fruit doesn't exist. Additionally, since ITP is built based on task guidelines alone, it demands significantly less prompt engineering than Code as Policies, which makes our system very easy to use for various task planning purposes.

## 4.3 Replan with Human-in-the-loop Feedback

Our system is robust to diverse new requests during execution. To verify this point, we assess the task replanning performance on real robots in response to a user's new request, referred to as human-in-the-loop feedback. We display the results in Table 2. We notice that ITP demonstrates its capacity to effectively handle a range of new requests, even after progressing through various steps of the task. The last example is of particular note, where ITP adds one step more ('Stir the mixture until the matcha powder is well mixed') before putting the working cup in the finished location. Here the language model assumes the need to stir

| User Request | New Request | Step When New Request is Made | | |
|---|---|---|---|---|
| | | 1st | 2nd | 3rd |
| *Can I have a cup of strawberry milk?* | *I want to add boba into the drink.* | 4/4 | 3/3 | 5/5 |
| *I want a matcha latte.* | *Sorry, I want boba bilk without matcha instead.* | 3/3 | 5/5 | 5/5 |
| *May I have a cup of milk with taro?* | *Can I replace the taro with strawberry?* | 3/3 | 5/5 | 5/5 |
| *Can I get a strawberry boba milk .* | *Sorry, can I reorder a strawberry milk?* | 3/3 | 5/5 | 5/5 |
| *A strawberry matcha milk with boba.* | *Can I just get matcha boba milk and no strawberry?* | 4/4 | 5/4 | 7/6 |

Table 2: Replanning performance with real robots given human-in-the-loop feedback. After the user sends a request, we interrupt the procedure before different steps (1st, 2nd, and 3rd). Note that our replanning system is robust in handling these new requests. Interestingly, for the last example, after the 2nd and 3rd step, ITP adds one step more ('Stir the mixture until the matcha powder is well mixed') before putting the working cup in the finished location, leading to 5 and 7 steps instead of 4 and 6 steps respectively. We assume this is because GPT-4 assumes matcha powder is hard to mix, while we select water-soluble matcha powder. Including the instruction 'matcha powder is water-soluble' in the task guidelines could address this issue.

| Task Description | $|A|$ | ProgPrompt | ITP |
|---|---|---|---|
| *watch tv* | 3 | $0.42 \pm 0.13$ | $0.83 \pm 0.06$ |
| *turn off light* | 3 | $1.00 \pm 0.00$ | $0.75 \pm 0.00$ |
| *eat chips on the sofa 5* | 5 | $0.40 \pm 0.00$ | $0.96 \pm 0.05$ |
| *brush teeth* | 8 | $0.74 \pm 0.09$ | $0.86 \pm 0.12$ |
| *throw away apple* | 8 | $1.00 \pm 0.00$ | $1.00 \pm 0.00$ |
| *make toast* | 8 | $1.00 \pm 0.00$ | $0.59 \pm 0.16$ |
| *put salmon in the fridge* | 8 | $1.00 \pm 0.00$ | $1.00 \pm 0.00$ |
| *bring coffeepot and cupcake to the coffee table* | 8 | $1.00 \pm 0.00$ | $1.00 \pm 0.00$ |
| *microwave salmon* | 11 | $0.76 \pm 0.13$ | $0.89 \pm 0.09$ |
| *wash the plate* | 18 | $0.97 \pm 0.04$ | $0.95 \pm 0.01$ |
| Avg: $0 \leq |A| \leq 5$ | | $0.61 \pm 0.29$ | $\mathbf{0.84 \pm 0.10}$ |
| Avg: $6 \leq |A| \leq 10$ | | $\mathbf{0.95 \pm 0.11}$ | $0.90 \pm 0.17$ |
| Avg: $10 \leq |A| \leq 18$ | | $0.87 \pm 0.14$ | $\mathbf{0.92 \pm 0.07}$ |

Table 3: Comparison of executability (Exec) on Simulation (Virtual Home) with ProgPrompt. Exec is the fraction of actions in the plan that are executable in the environment, even if they are not relevant for the task. ITP is not only a user-friendly approach that allows users to provide high-level guidelines, but it can also achieve superior results on varied tasks in the simulation.

the matcha due to the ambiguity of the correct procedure. Such superfluous steps can be reduced by adding restrictions in the task guidelines, which can easily be done by a general user of the system. This contrasts with methods like Code as Policies which require tuning prompts at the code level.

## 4.4 Comparison on Simulation Tasks

In this section, we aim to verify ITP's performance for simulation tasks. We compare our high-level planning module to that of ProgPrompt (Singh et al., 2023) by leveraging the simulated Virtual Home (VH) Environment (Puig et al., 2018). We would like to emphasize that ITP provides a user-friendly approach for users to input their high-level guidelines, which requires little background knowledge, while other approaches such as ProgPrompt employ a code-like prompting strategy. To make a fair comparison, we obtain high-level planning from both ITP and ProgPrompt, and use ProgPrompt's low-level execution, strictly following the same evaluation protocol: each result is averaged over 5 runs in a single VH Environment across 10 different tasks. We display the results in Table 3. We notice that ITP is not only a user-friendly approach that allows users to provide high-level guidelines in more straightforward natural language but is able to match or exceed ProgPrompt on executable functions on the Virtual Home benchmark.

### 4.5 Adaptation to Other Tasks

ITP is simple to adapt to new tasks. The system is principally reliant on task guidelines during high-level planning and predefined functions during low-level execution. This structure negates the need for intricate code implementation examples, making the system's generalization to other tasks remarkably straightforward. Refer to Figure 3 for the necessary components that need to be adapted. For adapting to a new task, only the Task Guidelines and documentation for the provided Low-level Skills need to be modified. Optionally, the Prompts for the high and low-level planner can also be tuned.

Task Guidelines 2: Task Guidelines for the Dishwashing Task. Again, we follow the simple format from before. Possible task options are provided as well as their high level steps. Additional options and tasks specific details are also provided to help guide the LLM during planning.

```
Options:
Wash one plate with rose flavor,
Wash all the plates and there are two plates,
Wash one plate and one fork
Instructions:
Wash one plate with rose flavor
Material: rose detergent
Steps:
0) grasp the dirty plate
1) remove large particle from the plate
2) open the dishwasher
3) pull out the rack
4) put one plate on the third rack
5) add rose detergent into the detergent dispenser
6) close the dishwaster
7) select the cycle and start dishwasher
8) after the dishwasher cycle is complete and the
dishwasher has stopped, wait a few minutes for the dis
to cool down
9) make sure the plate is clean and dry, otherwise
go into step 8)
10) return the clean plate to the finished location
Wash all the plates and there are two plates
Material: original detergent
0) grasp the first dirty plate
1) remove large particle from the plate
2) open the dishwasher
3) pull out the rack
4) put the plate on the third rack
5) grasp the second dirty plate
6) remove large particle from the plate
7) put the plate on the third rack
8) add original detergent into the detergent
dispenser
9) close the dishwaster
10) select the cycle and start dishwasher
```

```
11) after the dishwasher cycle is complete and the
dishwasher has stopped, wait a few minutes for the
dishes to cool down
12) make sure the plate is clean and dry, otherwise
go into step 8)
13) return all clean utensils to the finished
location
Wash one plate and one fork
Material: original detergent
0) grasp the dirty plate
1) remove large particle from the plate
2) open the dishwasher
3) pull out the rack
4) put the plate on the third rack
5) grasp the fork
6) remove large particle from the fork
7) put the fork on the first rack
8) add original detergent into the detergent
dispenser
9) close the dishwaster
10) select the cycle and start dishwasher
11) after the dishwasher cycle is complete and the
dishwasher has stopped, wait a few minutes for the
dishes to cool down
12) make sure the plate and fork are clean and dry,
otherwise go into step 8)
13) return all clean utensils to the finished
location
Available location we have now:
* first rack for forks and small kitchen utensils
* second rack for bowl/cup
* third rack for plate/big kitchen utensils
Available material we have now:
rose detergent, original detergent
```

We provide an additional example of adapting to an additional task with ITP. We adapt our system to study the high-level task planning capabilities of the task dishwashing. We simply replace the task guidelines for 'making a drink' with 'dishwashing'. We show the dishwashing task guidelines below in Task Guidelines 4.5.

We evaluate the high-level planning capability on how many steps are generated correctly. We show our results in Table 4. We find out that ITP performs very well on the novel dishwashing task. It has the capability not only to produce precise and novel instructions for new objectives but also to exhibit resilience when faced with entirely different tasks.

### 4.6 Analysis and Discussion

**Analysis.** Although ITP demonstrates great generalization ability, it relies solely on LLMs for replanning. As a result, errors from the recognition system or robot execution could lead to task failure. For instance, if the recognition system mistakenly identifies milk as 'taro jam', the robot might prepare the wrong drink by using the incorrect ingredient. Similarly, if a line becomes entangled around the robot arm and knocks over a cup containing liquid, the task could fail.

| User Request | Task Type | High-level Planning |
|---|---|---|
| *Wash one dirty plate with rose flavor.* | Existed | 11/11 |
| *Please wash 1 dirty bowl with rose flavor.* | Zero-shot easy | 11/11 |
| *Please clean the 2 dirty cups.* | Zero-shot easy | 14/14 |
| *Wash all forks, there are 3.* | Zero-shot easy | 17/17 |
| *Can you wash 2 plates? (New request: Can you wash another?)* | Zero-shot easy | 17/17 |
| *Please wash 2 forks and one bowl.* | Zero-shot moderate | 17/17 |
| *May you wash 2 cups and 2 plates?* | Zero-shot moderate | 20/20 |
| *Please wash 2 fork, 2 plate and 2 bowl.* | Zero-shot hard | 27/27 |
| *Wash 2 plates,1 bowl, 1 fork and 1 knife with rose flavor.* | Zero-shot hard | 23/23 |
| *Wash one dirty plate with lemon flavor* | Unavailable material | - |
| Total | - | 100% |

Table 4: Generalization to dishwashing task. We only need to change the text guidelines to make an accurate high-level plan. Since using the dishwasher to clean the dishes doesn't contain misleading material or content, the high-level planning rate is 100%. Please note that different utensils should be placed in different locations in the dishwasher, while ITP remains resilient in generating precise plans for each step, ensuring the correct order and appropriate location for different utensils. We envision the versatility of ITP's capabilities being applicable to a wide range of tasks.

To address these challenges, we can design a replanning system using vision-language models (VLMs). VLMs can detect potential dangers in the system and generate actions to mitigate these issues. Furthermore, due to the latency of several seconds when using GPT-4, tasks in our experiments sometimes take longer to complete. However, this issue can be alleviated by adopting open-source models, such as Llama (Dubey et al., 2024), as advancements in these techniques are progressing rapidly.

**Discussion.** Although many works (Singh et al., 2023; Liang et al., 2022; Skreta et al., 2023; Rana et al., 2023) have explored LLMs for understanding feedback and planning, none of these works both consider user-friendly task guidelines and replan based on a user's new request. Progprompt (Singh et al., 2023) and Code as Policies focus more on making one plan and executing the task step by step, while also requiring complicated reference code. CLAIRIFY (Skreta et al., 2023) provides effective guidance to the language model by generating structured task plans and incorporating any errors as feedback, and SayPlan (Rana et al., 2023) introduces an iterative replanning pipeline that refines the initial plan using feedback from a scene graph simulator. However, these two approaches pay attention more on a task instead of users' experience. Our ITP aims to provide a unified vision and language framework that can provide the best user experience, so the user with minimal specialized knowledge can still easily ask their robots to execute a task.

# 5 Conclusion

**Conclusion.** In this paper, we propose a simple yet effective system, ITP, which melds the capabilities of Large Language Models in an interactive robot system that constructs plans, and performs tasks centered around the users needs. Encouragingly, it precisely interprets user requests, generates pertinent step-by-step plans, and achieves the desired outcome — a testament to the potential of such systems for real-world applications. We embody our system in a robot designed to make various drinks according to user preferences and adeptly demonstrate its ability to respond to feedback during execution. Our system is capable in the context of interactive task planning and replanning for robotics.

**Limitations and Future Work.** While ITP provides a working proof of concept of an interactive robot system, there is room for enhancing its capabilities with more powerful robot skills to tackle more intricate tasks. Similarly, the integration of more precise visual information that leverages 3D information would significantly elevate the robot's proficiency in understanding, planning, and interacting with its surroundings. We aim for our open-source system to inspire more research into using both established and emerging models to enhance real-world robotics.

## 6 Acknowledgement

Philipp Wu was supported in part by the NSF Graduate Research Fellowship Program. We thank the Machine Common Sense project and ONR MURI award number N00014-21-1-2801. We thank Valts Blukis for the insightful and valuable discussion. Pieter Abbeel holds concurrent appointments as a Professor at UC Berkeley and as an Amazon Scholar. This paper describes work performed at UC Berkeley and is not associated with Amazon.

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
