# OpenReview forum: "Interactive Task Planning with Language Models"
_TMLR — Accepted by TMLR_

### Review · Reviewer_aatG · 2024-08-16

**Summary Of Contributions:**

This paper proposes an interactive task planning framework leveraging GPT4 and tests it on a real milk tea making tasks as well as a virtual home environment.

Given a user query, task guidelines giving few show examples and a scene representation coming from a vision language model, the system produces a step by step high level plan. For each step, the high level step is turned into a sequence of low level primitives by a LLM based on the scene representation. Replanning is supported after each step by simply appending the new user query to the interaction history and generating a new high level plan.

The method is evaluated on a real world milk tea making task and a virtual home environment and outperfoms a code-as-policies baseline both on tasks covered by the guidelines and unseen queries.

**Audience:**

Yes

**Broader Impact Concerns:**

none.

**Claims And Evidence:**

Yes

**Requested Changes:**

Questions:
* I don't understand how exactly the low level skills are selected? Are these conditioned by the LLM based on the context and scene representation or are they completely fix? This could be clearer in the paper
* What happens if a new user query requires starting from scratch? What happens if a low level skill fails?

minor:
* please change to citations in parenthesis: e.g. SayCan Zeng et al. (2022); -> SayCan (Zeng et al. (2022));

**Strengths And Weaknesses:**

Strengths:
* simple effective system that performs well
* real world and sim evaluation

Weaknesses:
* how exactly low level skills are selected and conditioned seems a little unclear.
* ideally there would be more trials in real to get a sense of the robust of the system
* More analysis of exactly why this system outperforms baselines would be beneficial.

---

> ### Author Response · Authors · 2024-08-19
> **Response to Reviewer aatG**
>
> We would like to thank Reviewer aatG for taking the time to review our paper and provide valuable feedback, as well as for the quick review! We are pleased that the reviewer found our system to be "simple and effective" and appreciated our "real-world and simulation evaluation." We are happy to have the opportunity to address the reviewer’s questions and concerns.
>
> Weaknesses:
>
>
> **Q1. how exactly low level skills are selected and conditioned seems a little unclear.**
>
> Thank you for your question! Our low-level skills are pre-defined and hard-coded by us. The strength of ITP is that, after designing the high-level steps based on the user’s request, it selects each low-level skill from a pre-defined skill pool and executes each step individually. We apologize for any confusion and will clarify this in our final version.
>
> **Q2. ideally there would be more trials in real to get a sense of the robust of the system.**
>
> Thank you for your valuable suggestion! Since ITP focuses primarily on planning, we have tested its performance in both real-world scenarios and simulations. We have found that ITP performs robustly in interactive task planning. We will follow your recommendations and conduct more trials on other tasks.
>
> **Q3. More analysis of exactly why this system outperforms baselines would be beneficial.**
>
> Thank you for your suggestion. We would like to highlight that the superior performance of ITP is due to two key reasons: ITP is the first to propose a replanning pipeline that responds to a user’s new request (please refer to Figure 3 in our paper for details). Previous baselines, such as ProgPrompt and SayCan, only considered one-round execution and therefore could not handle replanning problems. ITP’s pipeline leverages the advantages of large language models (LLMs) with the following process: user request $\rightarrow$ high-level plans $\rightarrow$ low-level plans $\rightarrow$ code. In contrast, previous baselines followed a different pipeline: user request $\rightarrow$ generated high-level code $\rightarrow$ low-level code, which is harder to generalize.
>
>
> Questions:
>
> **Q4. I don't understand how exactly the low level skills are selected? Are these conditioned by the LLM based on the context and scene representation or are they completely fix? This could be clearer in the paper.**
>
> Thank you for your question! We apologize for not making this clear earlier; here is a more detailed explanation. Please refer to the right side of Figure 3 in our paper. We first generate the high-level plan and then organize a prompt based on the completed steps and the high-level plan. This prompt is fed into a functional API (link: https://openai.com/index/function-calling-and-other-api-updates/), which enables GPT-4's function-calling capabilities to directly interact with the robot. The GPT-4 functional API will then search for the corresponding low-level plan code for execution. In the figure, GPT-4 returns grasp_empty_cup_from_stack(). Meanwhile, the scene representation provides the parameters for the location of each object. These parameters are dynamic and change based on the objects' locations. The robot then executes the step based on the code and the parameters derived from the scene description. We will include this explanation in our final version.
>
> **Q5. What happens if a new user query requires starting from scratch? What happens if a low-level skill fails?**
>
> Thank you for your question. We have provided answers to your questions below:
> 1) If a new user query requires starting from scratch, ITP will discard the existing cup and redo the experiments from the beginning. This is because ITP considers both the user’s new request and the steps already completed.
> 2) If a low-level skill fails, we did not include this in our submitted version. However, we could incorporate this feedback into ITP, allowing it to redo the step to ensure the task is completed. We will include this in our final version.
>
>
> minor:
>
> **Q6. please change to citations in parenthesis: e.g. SayCan Zeng et al. (2022); -> SayCan (Zeng et al. (2022));**
>
> Thank you for your valuable suggestion! We have updated the citations in parentheses. Please refer to the updated submission (PDF) for details. Please let us know if you have any further comments!
>
> We hope we could address your concerns satisfactorily. Please let us know if there are any new concerns or additional questions we can respond to.

---

### Review · Reviewer_LFrR · 2024-09-30

**Summary Of Contributions:**

The paper addresses the problem of  interactive robot learning for long-horizon task planning that can generalize to new goals or distinct tasks. The paper proposes ITP, where a high level planner takes as input a prompt to specify and a user request to output a step by step plan, followed by  a low level executor which tries to achieve a given step by converting robots skills into a functional API. The paper claims that, when the task changes, the approach is able to be replan accordingly with precision based on the user input, the task guidelines and previously executed steps. Overall the biggest strength of ITP is its capacity to effectively handle a range of new requests, even after progressing through various steps of the task.

Overall, I find the paper worthy of acceptance, and believe its merits outweighs its flaws. It is worthwhile to see the replanning abilities of large models with simply in-context learning. With this I recommend acceptance and encourage the authors to report the robustness in the results (i.e. how many seeds were used to test, what is the standard deviation across seeds?)  Please see weaknesses and requested changes.

**Audience:**

Yes

**Claims And Evidence:**

Yes

**Requested Changes:**

**Requested changes**

**See from weaknesses:**

5.   In Table 1: when you report “Quantitative results with real robots for high-level planning rate and success rate with various user requests.” How many seeds are these results over? Considering there is stochasticity in the LLM responses, how robust are the plans in meeting success criteria?
6.   While it is true that,  “since ITP is built based on task guidelines alone, it demands significantly less prompt engineering than Code as Policies, which makes our system very easy to use for various task planning purposes.” isn't the success of this method based on knowing the task ahead of time in the form of task guidelines, which significantly limits scaling this approach?


**Questions and clarifications**

Why do we not have a baseline performance reported in  Table 4: Generalization to dishwashing task. Could the authors please report those as well for baselines.

Task Guidelines 2 are quite comprehensive. What is the performance when you test generalization without providing such detailed task guidelines during zero-shot generalization?

What is missing from the manuscript is clarity and more transparency on how much time does it take for the system to replan? Considering there is no fine tuning, how are failure modes handled? Considering LLMs do hallucinate, it would be useful to add these modes in the manuscript.

**Strengths And Weaknesses:**

**Strengths:**

1.    ITP offers a real world system that has the ability of adaptive planning by means of interaction with the user. A  first plug-and-play LLM lists each step of a proposed plan. Each step of this plan is subsequently realized by another LLM with access to the functional API of the robot. A pre-trained VLM grounds the visual scene into language.
2.   The high level planner is technically the most straightforward approach which still lends itself to generalization - which is great to see functional in a real world system!
3.   The vision system design offers “a holistic ‘understanding’ of the scene, despite the location assignments being imprecise”.  Which potentially allows room for generalization.
4.   I believe the core strength and novelty in ITP is to consider new requests from the user as human-in-the-loop feedback wherein they allow humans to interpret the robot execution at any stage with a new prompt, which triggers the replanning pipeline which also considers the previous attempts etc.
5.   The method is compared with two baselines Code as Policies and ProgPrompt where the baseline is provided with the task guidelines same as IPT that include multiple instances, along with their associated high-level planning steps, current available material, and other task-specific conditions.


**Weaknesses:**
1.    Positioning of the work in the literature:  In Related work, authors mention: “In contrast to prior work, we allow the LLM to generate a high-level plan based on contextual information.” This is confusing because prior work also asks LLM to generate high level plans and very much based on contextual information. Can the authors please clarify the positioning of the work further?
2.   Assumption of a functional API of the robot: I believe this is a strong assumption and might be hard to generalize across different scenarios as that itself poses an open problem?
3.   The high level planning is basically a few-shot prompting mechanism which is not very novel given the abundance of work present along these lines.
4.   In this work, the  language model interfaces with a predefined skill set in Python that controls the robot.” What happens when the robot needs to update a skill when a new task is presented? What are the failure modes of this method since it's a training free method and adaptation is primarily by means of iterative prompting?
5.   In Table 1: when you report “Quantitative results with real robots for high-level planning rate and success rate with various user requests.” How many seeds are these results over? Considering there is stochasticity in the LLM responses, how robust are the plans in meeting success criteria?
6.   While it is true that,  “since ITP is built based on task guidelines alone, it demands significantly less prompt engineering than Code as Policies, which makes our system very easy to use for various task planning purposes.” isn't the success of this method based on knowing the task ahead of time in the form of task guidelines, which significantly limits scaling this approach?
7. Lack of clarity on failure modes, and details on real time adaptation- see requested changes.

---

> ### Author Response · Authors · 2024-10-10
> **Response to Reviewer LFrR (1/2)**
>
> We sincerely appreciate the reviewer LFrR’s time and thoughtful feedback on our paper. We are delighted that the reviewer recognizes the value of demonstrating functionality in a real-world system and sees potential for further generalization within ITP. We are happy to have the chance to address the reviewer’s questions and concerns.
>
> Weaknesses:
>
> **Q1. clarify the positioning of the work in related works.**
>
> We apologize for any misleading wording in the related works section. Our intention was to state that, unlike previous approaches, our work “designs text-only, user-friendly task guidelines, allowing the LLM to generate a readable high-level plan. For each high-level plan, the LLM directly obtains and executes low-level robot skills by accessing the robot's functional API, using a pre-trained VLM to ground the visual scene into primitives. Our work focuses on how to instantiate such a system in the real world.” We will update the text in the final version. Thank you for your valuable suggestion!
>
>
> **Q2. generalization ability across different scenarios.**
>
>
> Thank you for your question. Please allow us to provide a detailed explanation below.
> 1) Yes, this is a fair consideration. We acknowledge that this might be a strong assumption, but the functional APIs we used are developed based on existing LLMs. Given the strong generation capabilities of LLMs, we believe it wouldn't be an issue to design such a functional API.
> 2) Since the functional API is designed to search for low-level skills to complete high-level plans, different scenarios may share the same low-level skills but involve different objects. Therefore, we believe ITP has the capacity to generalize across various scenarios. This is similar to how humans perform tasks—learning a skill and applying it in different contexts. For instance, the functional API can call a 'put' function to 'put a plate on the table' in a kitchen for food preparation or 'put a cup on the table' in a dining room for pouring a drink.
>
>
> **Q3. The high-level planning is not novel.**
>
> Thank you for your comment. We apologize for any misleading wording in our related works (please also refer to our response to Q1 at your convenience). We do not claim high-level planning only as our contribution. Instead, our focus is on providing a real-world system capable of adaptive planning or replanning through interaction with the user. We see this as a key issue that hasn't received enough attention in previous research.
>
> **Q4. What happens when the robot needs to update a skill when a new task is presented? What are the failure modes of this method since it's a training free method and adaptation is primarily by means of iterative prompting?**
>
> We’d be happy to explain in detail:
> 1) If a new task is presented and is similar to a pouring task, we can directly call the existing low-level skill from the skill set via the functional API. However, if the new task is entirely different—for example, washing dishes—we should add the necessary low-level skill, such as 'wash()', to the skill set, while keeping the other steps unchanged.
> 2) Since daily tasks don't require complex planning, even though ITP is training-free, it can still achieve accurate high-level planning without the need for iterative prompting. The failure modes primarily arise from low-level control, which is a common issue across all robotic systems and not the main focus of our paper (please also refer to Q7 for an example of a failure mode at your convenience).
>
>
> **Q5. The number of running seeds in Table 1 and how robust are the plans in meeting success criteria?**
>
> Thank you for your questions. Please allow us to provide a detailed explanation below.
> 1) We take the fairness of our experiments seriously. For our results, we do not explicitly set a seed; instead, the system generates a random seed automatically, which varies with each run of the code.
> 2) While there is some stochasticity in the LLM responses, the LLM generates high-level plans based on our task guideline format, which provides a strong prior for the output. As a result, ITP remains highly robust for high-level planning and replanning (as demonstrated in Tables 1 and 4).

---

> > ### Author Response · Authors · 2024-10-10
> > **Response to Reviewer LFrR (2/2)**
> >
> > **Q6. Will task guidelines limit scaling this approach?**
> >
> > Thank you for your thoughtful question! 1) We would like to point out that previous approaches often rely on much more detailed task knowledge, such as pre-written code examples for specific tasks. Methods like SayCan or ProgPrompt directly generate new code using LLMs, but this requires users to provide code examples, and it can be difficult for the LLM to handle replanning. In contrast, ITP offers a simpler way for users to provide task guidelines. If the user wants to change their request, ITP only needs to replan at the high-level, then call and execute each low-level skill based on the new high-level plan. 2) As far as we are aware, task guidelines do not limit the scalability of this approach. LLMs can generalize across different scenarios for high-level planning, and the task guidelines merely provide a standard to ensure the LLM’s output follows the desired format.
> >
> > **Q7. Clarity on failure modes, and details on real time adaptation.**
> >
> > Thank you for your insightful comments.
> > 1) Regarding failure modes, ITP generally performs well in most cases, except for some specific situations. For example, our robot's gripper is wide, so when it grasps a cup, it may accidentally knock over a neighboring cup. This could be addressed by using a thinner gripper or incorporating a replanning module to better control low-level skill execution.
> > 2) There may be a slight delay in real-time adaptation due to the latency of using GPT-4, with a lag of several seconds. This can be mitigated by using an in-house LLM, such as Llama, to enable real-time adaptation.
> >
> > **Requested changes:**
> > Please review our responses to Q5 and Q6. We will incorporate all the relevant information in the final version of our paper. Thank you for your valuable feedback, which has helped strengthen ITP!
> >
> > Questions and clarifications:
> >
> > **Q8. Baseline performance reported in Table 4**
> >
> > Thank you for your question. In Table 4, our primary aim is to demonstrate the new capability of ITP - replanning. However, the previous baselines were not designed for replanning and do not include a dishwashing setup. Implementing additional code would be necessary to adapt these previous methods for new tasks, which may not provide a fair comparison.
> >
> > **Q9. Zero-shot Generalization?**
> >
> > Thank you for your valuable suggestion! One advantage of ITP is that high-level planning relies solely on text, without the need for code generation. As a result, even with limited guidelines, ITP can still produce effective high-level plans. For instance, in Task Guidelines 2, if we remove the first two examples and only retain the last example, “Wash one plate and one fork,” ITP is still able to achieve a 100% high-level planning success rate.
> >
> > **Q10. Time for replanning and solutions for failure modes.**
> >
> > Thank you for your thoughtful consideration.
> > 1) Due to the latency associated with using GPT-4, there may be several seconds of delay in real-time adaptation. This can be mitigated by employing an in-house LLM, such as Llama, to achieve real-time adaptation.
> > 2) Regarding failure modes, ITP generally performs well in most cases, except for certain specific situations. For example, our robot's gripper is wide, which can cause it to accidentally push down a neighboring cup when grasping one. This issue can be addressed by using a thinner gripper or by implementing a replanning module to control low-level skill execution.
> > Thank you for your valuable suggestions! Indeed, it would be beneficial to include these failure modes in the manuscript, as they provide useful information for the replanning module in managing low-level skill execution.
> >
> > Thank you for your valuable suggestions to strengthen ITP! We hope we have addressed your concerns satisfactorily. Please let us know if you have any new concerns or additional questions that we can assist with.

---

### Review · Reviewer_VzSU · 2024-10-06

**Summary Of Contributions:**

This study employs a large language model (LLM) as the foundation for its language processing. The proposed system, ITP, comprises two main components: a high-level planner and a low-level executor. The high-level planner takes as input a given prompt, task guidelines, and a user request, and outputs a step-by-step plan to execute the request. A low level executor converts the given step using LLM function calling capabilities to directly interact with the robot, abstracting code level details from the system. The paper also showcases ITP’s ability by embodying a robot to make drinks according to user preferences and requests.

**Audience:**

Yes

**Claims And Evidence:**

Yes

**Requested Changes:**

The main request will be a better error analysis section where the authors can discern failure cases and why failure occurred. Which part of planning and executor mainly is responsible and possible path to improve it. This is not critical to the paper being accepted but it is also not just a nice to have. This is something that will improve the paper for me. It is something that brings the paper from a borderline/weak accept to accept category.

**Strengths And Weaknesses:**

Strengths:

-- Key difference from prior work: ITP allows users to provide instructions in a way that's easy to understand and also adjust its plan if the user makes a new request.

-- Their experiments where ITP is implemented on a robot designed to make a variety of drinks based on user requests. They effectively showcased ITPs ability to adapt to feedback and make changes while preparing the drinks.


Weakness / Questions:

-- What happens when there is a discrepancy between the task guideline and the scene? How does ITP handles it, is it in the first stage or the second?

-- How does error in DINO-bounding box, perception of location and error from GPT in planning compound i.e. are there scenarios where the errors add up on each other failing the task? A better error analysis of when failures occur and what to do for those cases will be helpful

-- If I understand correctly the task guidelines are complete and the scene only reflects the scenarios mentioned in the task guidelines. I am unsure about the generalization capabilities of the method. In the paper, generalization to new goals doesn’t necessarily mean to goals that are not explained in the guidelines. Please correct me if I am wrong.

---

> ### Author Response · Authors · 2024-10-10
> **Response to Reviewer VzSU**
>
> We would like to thank Reviewer VzSU for taking the time to review our paper and provide valuable feedback. We are pleased that the reviewer found our experiments to “effectively showcase ITP's ability to adapt to feedback and make changes while preparing the drinks.” We are excited to have the opportunity to address the reviewer’s questions and concerns.
>
> Weaknesses:
>
> **Q1. What happens when there is a discrepancy between the task guideline and the scene? How does ITP handle it, is it in the first stage or the second?**
>
> Thank you for your question. ITP will address the discrepancy between the task guideline and the scene in the first stage. Since LLMs are good at understanding and generalizing text, we believe that ITP can create accurate high-level plans using both the scene and task guidelines, even if there is a discrepancy.
>
> **Q2. Error from perception or GPT analysis of when failures occur.**
>
> Thank you for your valuable suggestion! Please allow us to provide a detailed explanation below.
> 1) Based on our experiments, Grounding-DINO is robust in detecting the location of each object. The only failure case occurs in very dim lighting, where Grounding-DINO may struggle to distinguish between the strawberry (red) and taro (purple) due to their similar colors. This is a common issue in robotics, which we can address by ensuring the environment has adequate lighting conditions. Alternatively, we could fine-tune Grounding-DINO to enhance its robustness against varying light conditions.
> 2) Our experiments indicate that GPT planning can be reliable, as generating steps to accomplish tasks based on existing examples is not particularly challenging for LLMs. This has been verified by various experiments across different tasks, as shown in Tables 1, 2, 3, and 4.
>
> **Q3. Generalization ability based on the task guidelines.**
>
> Thank you for your question; there may be some misunderstandings. Please allow us to explain in detail. Task guidelines are intended to provide examples that control the output format of high-level steps, ensuring alignment with the users’ needs (as reflected in the task guidelines). For instance, in Task Guidelines 1, the user provides guidelines for making milk, strawberry milk, and boba milk. However, ITP is capable of generating high-level plans for any drinks, including but not limited to boba taro milk and matcha latte.
>
> **Requested Changes:**
> We sincerely appreciate your suggestions! Indeed, we will add an error analysis section in our final version to discern failure cases and why failure occurred as well as the possible solutions. In summary, the primary failure cases will come from three parts: 1) error from perception - which can be avoided by using advanced Vision-language model or setting the environment under a good condition; 2) latency from GPT (as GPT might take a few seconds to respond and generate plans) - this can be alleviated by using in-house LLMs such as Llama instead of cloud-based API such as GPT to largely reduced the latency; 3) ITP generally works very well for most cases unless some special case. For example, our robot gripper is wide, so when it grasp a cup, it might push another neighboring cup down - this can be avoided by using a thinner gripper, or setting up a replanning module to control the low-level skill execution. We will follow your instructions and add a specific section in our final version.
>
> We sincerely appreciate your suggestions! We will add an error analysis section in our final version to identify failure cases, explain why they happen, and suggest possible solutions. In summary, the main failure cases will come from three areas:
> 1) Perception errors: These can be reduced by using better vision-language models or ensuring that the environment is in good condition for accurate detection.
> 2) Latency from GPT: GPT may take a few seconds to respond and create plans. We can solve this by using in-house models, like Llama, instead of cloud-based APIs like GPT, which will help reduce the waiting and response time.
> 3) Specific failure situations: While ITP works well in most cases, some situations may cause issues. For example, our robot’s wide gripper might push down a neighboring cup when trying to grab another one. We can avoid this by using a thinner gripper or setting up a replanning module to manage low-level tasks. We will follow your advice and include this information in our final version.
>
> Thank you once again for your valuable suggestions to strengthen our paper on ITP! We hope we have addressed your concerns satisfactorily. Please let us know if you have any further questions or additional feedback.

---

### Decision · Action_Editor_sGiy · 2024-12-21

**Recommendation:** Accept with minor revision

**Comment:**

The core strength of the paper is we summarized in this review: "to consider new requests from the user as human-in-the-loop feedback wherein they allow humans to interpret the robot execution at any stage with a new prompt, which triggers the replanning pipeline which also considers the previous attempts etc."  The proposed approach develops a practical system that has the ability of adaptive planning by means of interaction with the user.  The method is compared with two baselines: Code as Policies and ProgPrompt, and shows favorable comparisons.

The main shortcomings raised were lack of clarity on failure modes of the system, and somewhat weak novelty on the overall architecture. However, the reviews overall are positive for the contributions of this paper.

**Audience:**

Yes, the problems studied in the paper are central to agentic AI systems including Robotics.

**Claims And Evidence:**

This paper explores replanning abilities of LLMs with in-context learning. The paper assumes a high-level LLM planner engaging with a low-level executor. Such an architecture is arguably standard by now in various application contexts, but the feedback aspect of humans-in-the-loop explored here is novel.  The approach is natural and the empirical evidence seems sufficient. The reviewers have requested a stronger error/failure analysis section clarifying responsibility of planning and execution; and more details on the latency characteristics of the whole system.